# Toolkit for incoherent scatter radar experiment design and applications to EISCAT\_3D

Spencer Mark Hatch<sup>1</sup>, Ilkka Virtanen<sup>2</sup>, Karl Magnus Laundal<sup>1</sup>, Habtamu Wubie Tesfaw<sup>2</sup>, Juha Vierinen<sup>3</sup>, Devin Ray Huyghebaert<sup>4</sup>, Andres Spicher<sup>3</sup>, and Jens Christian Hessen<sup>1</sup>

**Correspondence:** Spencer M. Hatch (spencer.hatch@uib.no)

## Abstract.

Modern phased-array incoherent scatter radar (ISR) systems consist of several thousand phased-array antenna elements. Next-generation phased-array ISR systems are shifting towards multistatic setups consisting of three sites, such as EISCAT 3D with sites in Finland, Norway, and Sweden. The tremendous flexibility that these ISR systems afford also presents a challenge: Given a science question and an estimate of the associated ionospheric conditions, how does one begin to design an ISR experiment? Here we present a method for performing observing system simulation experiments (OSSEs) with multistatic and monostatic ISRs. The method estimates the variance, or uncertainty, of measurements of three scalar quantities (plasma density, electron and ion temperature), and the covariance of one vector quantity (ion drift) in the case of multistatic systems. It is based on analytic first-order linearization of the incoherent scatter spectrum, as well as inverse and radar theory. Uncertainty estimation requires specification of the radar system as well as plasma density, electron and ion temperature, ion-neutral collision frequency, the fractional density of O<sup>+</sup>, and the mean molecular mass. We validate this analytic uncertainty estimation method against uncertainty estimates derived directly from EISCAT incoherent scatter radar measurements made over Tromsø. We also present an open-source implementation of this method and additional tools written in R and Python that may be used to assess whether a candidate experiment is likely to achieve the temporal and spatial resolution needed to study a particular phenomenon. The user may vary parameters such as integration time, bit length, and duty cycle to understand their effect on experimental uncertainties. By default the EISCAT 3D radar configuration is used and these parameters are calculated automatically via two commonly used empirical models; it is nevertheless straightforward to manually specify alternative radar configurations, whether mono- or multistatic, and individual ionospheric and atmospheric parameters. We show how different beam patterns affect reconstruction of the ionospheric potential electric field, and present an example experiment optimized for reconstructing the electrodynamics around an auroral arc.

<sup>&</sup>lt;sup>1</sup>Department of Physics and Technology, University of Bergen, Bergen, Norway

<sup>&</sup>lt;sup>2</sup>Space Physics and Astronomy Research Unit, University of Oulu, Oulu, Finland

<sup>&</sup>lt;sup>3</sup>Institute for Physics and Technology, University of Tromsø, Tromsø, Norway

<sup>&</sup>lt;sup>4</sup>Leibniz Institute of Atmospheric Physics at the University of Rostock, Kühlungsborn, Germany

## 1 Introduction

Incoherent scatter radar (ISR) is an active remote sensing technique for estimating plasma parameters – most often plasma density, plasma drift, ion temperature, and electron temperature, but it is also possible to estimate or infer ion compositions (e.g., Kelly and Wickwar, 1981; Lathuillere et al., 1983a; Häggström and Collis, 1990; Cabrit and Kofman, 1997; Zettergren et al., 2010; Blelly et al., 2010; Virtanen et al., 2024), ion-neutral collision frequencies (e.g., Dougherty and Farley, 1963; Oyama et al., 2012; Lathuillere et al., 1983b; Nicolls et al., 2014; Günzkofer et al., 2023) and/or neutral winds (Brekke et al., 1973; Heinselman and Nicolls, 2008; Nygrén et al., 2011; Stamm et al., 2021a). All of these quantities can be estimated via the so-called "ion line" of the incoherent scatter spectrum. This line is the product of electromagnetic radiation scattering off ion acoustic waves. Although much more challenging to measure and therefore much less frequently used, the "plasma line" can also be used in some situations to calibrate plasma density and electron temperature measurements, and estimate ion composition, electron drift velocity, and ion-neutral collision frequency (Akbari et al., 2017, and references therein).

The ISR technique is fundamentally governed by the radar equation. Given the radar cross section per electron ( $\sim 10^{-28}$  m<sup>2</sup>) and plasma densities of  $10^9 - 10^{12}$  m<sup>-3</sup>, the return power is a vanishing fraction ( $\sim 10^{-20}$ ) of the transmitted power. Incoherent scatter observations thus require high transmitted powers, high-gain antennas, and high transmission duty cycles. Large, steerable dishes comprise the bulk of existing ISR infrastructure worldwide. However, since the advent of the Poker Flat ISR (Nicolls et al., 2007), electronically steerable phased-array radars consisting of fields of smaller antennas have become the standard for modern ISRs. These next-generation radars have the advantage of being much more flexible in terms of beam steering (including post-experiment beam steering) and radar imaging.

The most recent generation of ISRs, including the low-latitude Sanya radar (He et al., 2020; Yue et al., 2022) and the high-latitude EISCAT\_3D (hereafter E3D) radar (Lehtinen et al., 2014), are phased-array systems and are ultimately designed to be multistatic. Multistatic radar systems can enable, for example, measurement of the full ion drift velocity vector for favorably located transmitter-receiver pairs (e.g., McKay-Bukowski et al., 2015; Virtanen et al., 2014). Multibeam remote receiver data can also significantly improve statistical accuracy of the plasma parameter observations via relatively independent measurements of the monostatic transmit beam at different altitudes from the remote sites, as compared to monostatic measurements in isolation (Virtanen et al., 2014). When completed, E3D will be a multistatic system with a transceiver in Skibotn, Norway (69.34°N, 20.31°E), and receivers in Kaiseniemi, Sweden (68.27°N, 19.45°E) and Karesuvanto, Finland (68.48°N, 22.52°E).

E3D represents the first attempt to exploit ISR techniques together with phased-array radar technology and multiple receiver sites at high latitudes to yield 3D measurements of three scalar quantities (plasma density and electron and ion temperatures) on timescales of seconds, and a 3D vector quantity (ion drift velocity) on timescales of minutes, within a large volume of order  $10^6 \text{ km}^3$ . E3D measurements therefore present an unprecedented opportunity to probe the true 3D structure of the ionosphere-thermosphere (IT) system on relatively short timescales.

Figure 1 displays the location of the three E3D sites as well as how they are situated relative to the thirteenth International Geomagnetic Reference Field (IGRF Alken et al., 2021) model for Earth's main magnetic field. Figure 1a shows the locations of the E3D sites in both geographic and geomagnetic (Modified Apex-110) coordinates. Figure 1b shows the angle between

the line-of-sight vector at the Skibotn transceiver and the main magnetic field, which reaches a minimum just south of Skibotn. Figures 1c and 1d respectively show contours of constant magnetic field strength and inclination at a reference altitude of 200 km.

E3D has the potential to perform unprecedented measurements of a wide variety of high-latitude phenomena (McCrea et al., 2015), including making volumetric estimates of the ion drift and neutral wind velocity vector fields. Using these measurements, it may be possible to make volumetric estimates of ionospheric perpendicular (relative to the background magnetic field) and even parallel current vectors (Reistad et al., 2024). Each of the 109 panels in the main cluster at Skibotn consists of 91 antennas. With the planned addition of 10 outlier panels in addition to the main cluster, it may also be possible to image the ionosphere at sub-beam spatial resolutions of less than 100 m (Stamm et al., 2021b; Huyghebaert et al., 2025).

Optimizing the science return of E3D and other, similar radar systems requires a flexible set of software tools for designing experiments that do not require the radar system to be turned on. These tools should ideally be sufficiently advanced to produce believable estimates of measurements uncertainties given a realistic radar configuration specified in terms of radar geometry, range resolution, and integration times.

Given such tremendous flexibility, where does one begin in designing an experiment? This study is motivated by the need for simple, flexible, open-source tools for designing a candidate E3D experiment, and assessing whether the experiment is likely to meet the observational requirements for answering a user-specified science question. One existing candidate would be the open-source tools presented by Swoboda et al. (2017), which are capable of simulating the full ISR analysis chain, including raw voltages. The Swoboda et al. (2017) toolbox nevertheless requires a relatively advanced understanding of ISR analysis and is somewhat computationally demanding; it is therefore most accessible to expert users. A more recent example specific to E3D has also been presented by Virtanen et al. (2025). We view the present work as being most useful to the researcher who does not need access to low-level data such as complex voltages and lag products, but is most interested in high-level data: plasma density, electron and ion temperatures, and ion drift velocity estimates. We assume that the user of these tools has in mind a particular scientific question and a target phenomenon or set of phenomena, defined in terms of temporal and spatial scales and order-of-magnitude estimates of ionospheric plasma and ion drift parameters. This study also assumes a basic grasp of concepts such as duty cycle, bit length and beam width (although the latter two are defined and discussed in Section 4.1).

The study is divided into the following sections: a description of the uncertainty estimation procedure (Section 2); implementation and validation of the uncertainty estimation procedure (Section 3); a general framework for designing an experiment (Section 4); two example experiments (Section 5); and discussion and summary (Section 6).

## 2 Estimation of plasma parameter uncertainties

60

80

The uncertainty estimation procedure that we use is based on a modified version of the procedure first presented in Appendix B of Lehtinen et al. (2014) that was developed by one of us (I. Virtanen). It generalizes the Vallinkoski (1988) method by accounting for non-zero ion drift and multistatic radar systems. However, it accounts neither for the possibility of anisotropic ion and electron temperatures nor for the phenomenon of Faraday rotation, as do more sophisticated procedures such as the

**Figure 1.** Summary of magnetic and geodetic coordinates, magnetic field strength and inclination at 200-km altitude, and E3D viewing angle relative to the background magnetic field at 200-km altitude. In each panel the site of the Skibotn transceiver (yellow star) and the receivers in Kaiseniemi, Sweden and Karesuvanto, Finland (orange stars) are shown. a) Modified Apex-110 coordinates (black lines) and rings indicating the field of view at Skibotn down to 30° elevation for 100-, 300-, and 500-km altitude (blue lines). b) Angle in degrees between Skibotn line of sight and unit vector in the direction of the IGRF magnetic field (reference date Dec 1, 2020). c) IGRF field strength. d) IGRF field inclination.

4

multi-static parameter and uncertainty estimation procedure presented by Virtanen et al. (2014). The procedure consists of two steps. Signal and noise powers are first calculated using the radar equation and user-defined system noise temperature, and the resulting ACF noise level is then used to calculate the plasma parameter error estimates.

## 2.1 Signal and noise powers

100

105

For a known radar system and radar operation mode, the signal power dP from a plasma volume dV at the point  $\mathbf{r}_t$  is

$$dP = \frac{N_e \xi_0 \sin^2 \chi P_T}{1 + T_e/T_i} \frac{G_T}{4\pi r_T^2} \frac{G_R}{4\pi r_R^2} \frac{\lambda^2}{4\pi} W_r dV, \tag{1}$$

where  $N_e$  is electron number density,  $\xi_0=10^{-28}~{\rm m}^2$  is the single-electron radar cross-section,  $\chi$  is the polarization angle (angle between the electric field vector of the incident wave and wave vector of the scattered wave),  $P_T$  is the transmitted power,  $T_e$  and  $T_i$  are electron and ion temperatures, respectively,  $G_T$  and  $G_R$  are gains of the receiver and transmitter antennas at  ${\bf r}_t$ ,  $r_T$  and  $r_R$  are distances from the target to the transmitter and the receiver, respectively, and  $\lambda$  is wavelength of the radar carrier signal.  $W_r$  is the range ambiguity function, which includes the effects of phase-coding and decoding in the signal processing chain, which narrows down the plasma volume in the range direction according to the applied coding and decoding scheme. The radar is assumed to transmit circular polarization and the receiver polarisation is assumed to be matched with the polarization ellipse of the scattered wave. Total power received from a scattering volume V with known  $N_e$ ,  $T_e$ , and  $T_i$  located around  ${\bf r}_t$  can be integrated using (1),

$$P = \iiint_V dP,\tag{2}$$

if technical specifications of the radar system and range resolution of the measurement are known. A detailed discussion of determination of the scattering volume V is provided in Appendix B of Lehtinen et al. (2014). To simplify and increase the computational speed of the calculations, the beam patterns and pulse shapes are assumed Gaussian.

Noise contributions from thermal noise in the radar receiver and sky-noise are described by the system noise temperature  $T_{sys}$ . For a known receiver bandwidth  $\delta\nu$  the background noise power is

$$P_B = k_B T_{sys} \delta \nu, \tag{3}$$

where k<sub>B</sub> is the Boltzmann constant. Another noise contribution, the so-called self-noise, arises from the phase-code decoding, which effectively converts the signal from unwanted ranges into zero-mean noise. The self-noise power P<sub>S</sub> can be integrated from (1) if the range ambiguity function W<sub>r</sub>, corresponding to one bit in the phase-coded pulse, is replaced with W<sub>p</sub>, matched to the length of the whole pulse. To get a reasonable and computationally light-weight approximation of the self-noise power, the self-noise calculation assumes that the electron density altitude profile has a Gaussian shape that peaks at r<sub>t</sub> and has a user-defined width. This approximation is used only in the self-noise calculation, not for the signal powers.

These approximations allow us to calculate the total noise power  $P_N = P_B + P_S$  and the signal-to-noise ratio as

$$SNR = \frac{P}{P_B + P_S}.$$
 (4)

One should note that the self-noise contribution  $P_S$  will often be significant in E3D due to its high transmitted power and large antenna gains. Ignoring the self-noise contribution would thus lead to unrealistically high SNR especially in the E region, where high electron densities are often present at a short distance from the radar.

## 2.2 Linearization of the incoherent scatter theory

The uncertainty estimation procedure relies on linearization of the non-linear incoherent scatter theory, which yields a relation between the incoherent scatter autocorrelation function (ACF) and the plasma parameters. We denote the theoretical ACF by  $f(\mathbf{k}, \tau; \mathbf{x})$ . Via the Wiener-Khinchin theorem the ACF is the Fourier transform of the incoherent scatter spectrum  $S(\mathbf{k}, \omega)$  given in Appendix A. The three arguments of the ACF are the scattering wave vector  $\mathbf{k}$ , the time lag  $\tau$  and the model vector of (normalized) plasma parameters

$$\mathbf{x} = (n_e/n_0, T_i/T_0, T_e/T_i, \nu_{in}/\nu_0, p, v_s/v_{s0})^T,$$
(5)

where  $n_e, T_i, T_e, \nu_{in}, p$ , and  $v_s$  are respectively the plasma density, ion temperature, electron temperature, ion-neutral collision frequency for momentum exchange, the fractional abundance of oxygen ions (assuming the plasma is composed of a mixture of NO<sup>+</sup>, O<sub>2</sub><sup>+</sup>, and O<sup>+</sup>), and the line-of-sight ion drift measured from site s. The variables  $n_0, T_0, \nu_0$ , and  $v_{s0}$  are normalization constants, and NO<sup>+</sup> and O<sub>2</sub><sup>+</sup> are represented as one molecular ion with mass 30.5 u. Thus the set of N measurements of the ACF at time lag  $\tau$  with scattering vector  $\mathbf{k}$  is represented by the measurement vector

$$\mathbf{m} = (d_0, d_1, \dots, d_N)^T, \tag{6}$$

which is related to the ACF via

135 
$$\mathbf{m} = f(\mathbf{k}, \tau; \mathbf{x}) + \epsilon.$$
 (7)

Here  $\epsilon$  is taken to be Gaussian zero-mean noise. For non-zero  $v_s$  the measurement vector m is complex.

In linearizing the ACF,

$$f(\mathbf{k}, \tau; \mathbf{x}^0 + \Delta \mathbf{x}) \approx f(\mathbf{k}, \tau; \mathbf{x}^0) + \mathbf{A} \Delta \mathbf{x},$$
 (8)

we define a theory matrix A that consists of the partial derivatives of f with respect to each plasma parameter  $x_k$ . As with the measurement vector  $\mathbf{m}$ , this matrix is complex for non-zero  $v_s$ ; it is nevertheless possible to write the real and imaginary parts of Equation (8) separately such that it may be represented as a purely real system of equations. Provided an estimate of the measurement noise covariance  $\Sigma_m$ , one can also calculate the posterior model covariance  $\Sigma_x$ , or covariance of plasma parameter estimates, using the standard result from linear inverse theory:

$$\Sigma_x = \left(\Sigma_{x,pr}^{-1} + \mathbf{A}^T \Sigma_m^{-1} \mathbf{A}\right)^{-1},\tag{9}$$

where  $\Sigma_{x,pr}$  is the prior model covariance matrix that represents any prior information that we may have about the distributions and interrelationships of the plasma parameters  $\mathbf{x}$ . (In this work we use  $\Sigma_{x,pr}^{-1} = 0$ .) Equation 9, which is the basis for uncertainty estimation, only requires specification of the theory matrix  $\mathbf{A}$  and does not require that  $\Delta \mathbf{x}$  be nonzero.

# 2.3 Uncertainty estimation procedure


At the highest level, our uncertainty estimation procedure consists of two steps: estimation of the ACF measurement error covariance matrix  $\Sigma_m$ , and plasma parameter uncertainty estimation via Equation 9.

In the first step we estimate the ACF measurement covariance matrix  $\Sigma_m$  as follows. We assume that measurements of the ACF at different lags are uncorrelated and have the same variance  $\sigma^2$ , so that  $\Sigma_m$  is an identity matrix multiplied by  $\sigma^2$ . With these assumptions, the ACF noise level may be defined as (Vallinkoski, 1988)

$$\gamma^2 = \frac{\sigma^2}{f(\mathbf{k}, 0; \mathbf{x})^2 N_\tau} = \frac{N_a}{N_\tau} \frac{1}{\text{SNR}^2},\tag{10}$$

with  $N_a$  the number of averaged ACF samples,  $N_{\tau}\approx 2\pi/(\omega_0\Delta\tau)$  the approximate number of lags to the first zero of the ACF assuming zero plasma bulk flow,  $\omega_0=k\sqrt{2k_BT_i/m_i}$  the "ion angular Doppler frequency" (Vallinkoski, 1988) for scattering wave number k, and  $\Delta\tau=2\Delta r/c$  the lag spacing (sampling interval) of ACF measurements. The lag spacing is matched with the user-specified range resolution  $\Delta r$ . The number  $N_a$  accounts for the number of samples averaged in phase-code decoding, as well as post-integration in range and time.

With this definition, we calculate the SNR and the ACF noise level at each site as described in Section 2.1 using the radar equation assuming a particular set of radar system parameters, and assuming Gaussian beam shapes and pulse shapes. (See Sections B.3–B.6 in Lehtinen et al., 2014, as well as the routine "multistaticNoiseLevels" in Virtanen, 2023.) The outputs from this step are the separate ACF noise levels  $\gamma_s$  for each transmitter-receiver pair s.

In the second step (plasma parameter uncertainty estimation) we use the fact that uncertainty of the line-of-sight velocity  $v_i$  is practically independent of the errors of the other parameters, and assume that all receivers see the same  $n_e$ ,  $T_e$ , and  $T_i$ . The latter assumption is equivalent to assuming that  $T_e$  and  $T_i$  are isotropic. (In reality  $T_e$  and  $T_i$  may be neither isotropic nor equal; see discussion in, e.g., Virtanen et al., 2014, and references therein.) The uncertainty estimation itself can therefore be divided into two parts: calculation of (i) the variance of estimates of scalar parameters, which do not have a direction, and (ii) the covariance of ion drift velocity vector estimates.

To calculate (i) we simply sum information from all receivers (since information or precision is the inverse of variance) to get a combined ACF noise level for the full multistatic system:  $\gamma = \left(\sum_s \gamma_s^{-2}\right)^{-1/2}$ . We then calculate the theoretical ACF  $f(\mathbf{k}, \tau = 0; \mathbf{x})$ , and solve for the ACF variance  $\sigma^2$  in Equation 10 (see routines "parameterFitErrors" and "IsspectrumSimple" in Virtanen, 2023). We thus have an estimate of the ACF covariance matrix  $\Sigma_m$ . We then calculate the theory matrix  $\mathbf{A}$ , and finally calculate the posterior model covariance matrix in Equation 9, with the scalar plasma parameter uncertainties taken to be the square roots of its diagonal elements.

To calculate (ii) one must consider the system geometry, and in general the estimate of the line-of-sight ion drift for each transmitter-receiver pair has a different uncertainty. We combine the different uncertainties following the methodology outlined in Appendix B. For a monostatic system the ion drift velocity covariance matrix reduces to a scalar representing the variance of the line-of-sight ion velocity.

## 3 Implementation and validation






We have produced an open-source toolkit consisting of a core implementation of the uncertainty estimation procedure described in Section 2 called ISgeometry (Virtanen, 2023), which is written in the R programming language, and a front-end called e3doubt (Hatch and Virtanen, 2024) that is written in Python. The latter includes scripts that reproduce the analysis described in this section and Section 5, as well as several additional tools. Exploration of different candidate beam patterns, for example, is straightforward and requires nothing more than a text-based list of beam pointing directions (elevations and azimuths).

To validate the uncertainty estimation procedure described in Section 2, we use field-aligned measurements made by the EIS-CAT Tromsø UHF radar during a geomagnetically quiescent period, 10-11 UT on March 3, 2023 at  $69.5864^{\circ}$  N,  $19.2272^{\circ}$  E  $(66.57^{\circ}$  magnetic latitude), equatorward of the geomagnetic cusp at magnetic local times between 11.8 and 12.8. During this period the y component of the interplanetary magnetic field in GSM coordinates hovered around 5 nT, while the z component was generally between -2 and 1 nT.

Figures 2a–d show that during this period, the temporal variability of the local ionosphere over Tromsø, in terms of  $n_e$ ,  $T_e$ ,  $T_i$ , and the line-of-sight ion drift velocity  $v_i$ , is minimal at each altitude. For the purpose of validation this low variability is desirable, because it permits us to compare three different estimates of the uncertainty of observed plasma parameters: (i) straightforward calculation of the standard deviation of each parameter at each altitude from the full hour of observations under consideration (time resolution of 1 min); (ii) uncertainty estimates provided by the standard GUISDAP analysis suite (Lehtinen and Huuskonen, 1996); (iii) the uncertainty estimation procedure that we describe in Section 2.

All three uncertainty estimates are shown in Figure 2e-h. For all parameters, the uncertainty estimates yielded by the procedure described in Section 2 (black dotted line labeled "e3doubt") are generally very close to the estimates provided by GUISDAP (thick gray line), with the deviation between the two increasing with increasing altitude. In contrast, the standard deviation of each parameter at each altitude that is calculated directly from the time series (thin orange line, labeled "sample") is greater than the other two estimates at virtually all altitudes. These differences apparently have to do with error correlations that are ignored both by standard GUISDAP analysis and the procedure presented in this study (Huuskonen and Lehtinen, 1996): One of us (IV) has found that the contribution of error correlations rather precisely accounts for the difference between GUISDAP uncertainty estimates and the sample standard deviations in a single test calculation. The inclusion of error correlations, however, raises the computational cost by a factor of  $10^4$  and is infeasible for routine calculations. (In the case of this study, "ignoring error correlations" refers to our assumption in Section 2.3 that the ACF covariance matrix  $\Sigma_m$  is an identity matrix multiplied by a constant.)

The overall similarity in Figures 2e—h between the uncertainty estimates given by GUISDAP and our uncertainty estimation procedure (e3doubt) indicates that both produce reasonable, if optimistic, estimates of the uncertainties of plasma parameters derived from ISR measurements. Beyond this main observation, some additional observations are in order. First, comparison of GUISDAP and e3doubt uncertainties points to an apparent tendency for e3doubt to underestimate uncertainties relative to those of GUISDAP above ~300-km altitude. Whether this difference is related to one or more of the assumptions we make in

Figure 2. EISCAT Tromsø ultra-high frequency (UHF) radar measurements made during 10–11 UT on March 3, 2023 (top four panels) and corresponding estimates of plasma parameter uncertainties based on three different approaches (bottom four panels). (a) Electron density  $n_e$ . (b) Electron temperature  $T_e$ . (c) Ion temperature  $T_i$ . (d) Line-of-sight ion velocity  $v_i$ . Panels (e)–(h) show uncertainties of  $n_e$ ,  $T_e$ , and  $T_i$  estimates based on (i) direct evaluation of the standard deviation of parameter estimates at each altitude for the hour during which observations were made (thin orange line); (ii) standard GUISDAP analysis (thick gray line); and (iii) the uncertainty estimation procedure described in Section 2 (black dotted line).

Section 2 is unfortunately unclear. On the other hand, the sample standard deviations indicated by orange lines in Figures 2e-h have slight local maxima at altitudes between  $\sim$ 225–275 km (e.g., local peaks in std( $n_e$ ) and std( $T_e$ ) near 250-km altitude). The e3doubt uncertainty estimates also exhibit local, albeit more subtle, maxima near 250-km altitude, whereas local maxima do not appear in the GUISDAP-based uncertainty estimates of  $n_e$  and  $T_e$ . The reason for these differences is again unclear.

# 4 Experiment design





Two of the first questions that are typically asked in discussions of E3D experiments are "How accurately will E3D be able to measure plasma parameters?" and "What will be the smallest possible spatial and temporal resolutions?" There is no universal answer to these questions, since accuracy and resolution depend intimately on the state of the ionosphere (cf. Equations 1, 5, and 8); ISR system properties such as radar geometry and self-noise (Equations 1–4 and 8); and experiment design choice such as range resolution, beam selection, and integration times (Equation 10), as outlined in Section 2.

In the following subsections we briefly discuss some general guidelines in experiment design (Section 4.1) and some considerations in the selection of bit length, integration time, and range resolution (Section 4.2). For a detailed discussion of ISR experiment design, see Chapter 2 in Lehtinen et al. (2014) and references therein.

## 4.1 Guidelines

A simple way to constrain the otherwise overwhelmingly large parameter space that comprises E3D experiment design is to begin with identifying a goal or set of goals that one wishes to achieve via a particular experiment, as is done in Section 5. These goals will often be related to a scientific hypothesis or question. In such cases, some general considerations for experiment design can be stated:

- (i) Gather information about the desired study targets or conditions. Where do they occur in terms of latitude, longitude, local time, and/or altitude? When do they occur in terms of time of day, season, or relevant solar wind or geomagnetic conditions? What are the typical spatial and temporal scales? Over what altitude range? Which plasma parameters are most important?
- (ii) On the basis of (i), identify requirements related to temporal and spatial coverage and temporal and spatial resolution.
- (iii) If possible, estimate acceptable levels of uncertainty in the plasma parameters (Equation 5).
- (iv) With knowledge of coverage, resolution, and/or uncertainty requirements, decide on possible beam patterns and bit lengths.
- Regarding the first point, the field of view of the Skibotn transceiver is confined to approximately 65–68° magnetic latitude (MLat) at 100-km altitude (indicated by the innermost blue ring in Figure 1, 68–71° geodetic latitude). Meanwhile, various auroral forms exist over different ranges of local times and MLats (e.g., illustration in Figure 1 of Knudsen et al., 2021). Thus whether phenomena located in the polar cap, auroral zone, or at sub-auroral latitudes can be observed by E3D is a question of

local time and geomagnetic activity. A detailed list of phenomena that may be observable with E3D has been given by McCrea et al. (2015).

## 4.2 Additional guidelines

# Bit length






The modulation bit length ( $\Delta\tau$  in Section 2) is an example of a parameter that needs to be carefully considered in experiment design, as it may have significant influence on the uncertainty of plasma parameter estimates. In a low-SNR setting, the bit length should be as long as possible (Lehtinen, 1989). This enables one to narrow down the receiver bandwidth and therefore minimize the background noise power. In the opposite case of high SNR, the bit length should be reduced to collect more samples per unit range, which improves the statistics (Lehtinen and Damtie, 2013). These considerations can present contradictory requirements for optimizing estimation of plasma parameters in the E region, F region, and topside ionosphere. In existing monostatic EISCAT experiments one finds bit lengths  $\Delta\tau=2$ –100  $\mu$ s, which correspond to range resolutions of  $\Delta r=0.3$ –15 km. The bit length must in general be equal to or shorter than the finest range resolution needed along a given beam.

## Range resolution

Range resolution is defined separately from the modulation bit length, because the ACF samples are routinely integrated in range to reach better statistical accuracy in those parts of the ionosphere where the best possible resolution is not needed. The range resolution can be freely selected, but a realistic choice is to use an integer multiple of the range resolution produced by the modulation bit length, because we assume that signal sampling is matched to the bit length. The resolution can vary along a beam such that one may opt, for example, for high resolution in the E region and coarser resolution in the F region. For a monostatic radar the range resolution can be converted to altitude resolution as  $\Delta h = \Delta r \cos(\text{elevation})$ , because range is measured along the radar beam. For a bistatic transmitter-receiver pair one must consider directions of both the transmitter and the receiver beams (Virtanen et al., 2014).

# **Integration time**

The integration time can in principle be selected for each beam individually, and it could be appropriate to use different integration times for different beams. A good starting point is nevertheless to use equal integration time for all beams to understand how parameter uncertainties vary from beam to beam. Then, to optimize the experiment, one may increase the integration time of beams where more precision is needed and decrease the integration for other beams where less precision is needed. For example, the geometry of the E3D sites is such that the uncertainty of measurements of some components of the ion drift vector increase with decreasing radar elevation; it may be appropriate in some cases to increase the integration time of beams with low elevations relative to those with high elevations.

One may also observe in Equation 10 the proportionality  $\sigma \propto N_a^{-1/2}$  between the ACF variance  $\sigma^2$  and the number  $N_a$  that accounts for the number of samples averaged in phase-code decoding and post-experiment integration in range and time. In other words, a sometimes useful rule of thumb is that the uncertainty of a particular plasma parameter estimate decreases by a factor of  $\sqrt{2}$  when the integration time is doubled.

## 5 Example experiments





In this section we present two examples demonstrating practical application of the uncertainty estimation framework presented in Sections 2–3.

For the first example we use the guidelines from Section 4.1 to design an experiment that is optimized for monitoring the properties of a discrete auroral arc and reconstruction of the surrounding plasma convection along a line of constant magnetic longitude during geomagnetically quiet periods.

For the second example we demonstrate how different beam patterns affect reconstruction of plasma convection within the same volume. This is accomplished by modifying the experiment originally presented in Reistad et al. (2024) for reconstruction of the ionospheric electric potential and plasma convection pattern.

## 5.1 Example experiment design #1: Quiet, discrete auroral arc

Auroral arcs and aurora-like features can appear at all local times within the latitudinal extent of the auroral oval, where the latter depends on the level of geomagnetic activity. A well known auroral form is the standard discrete arc that occurs during geomagnetically quiet periods – quiet, discrete aurora (hereafter QDA). In spite of its apparent simplicity, basic facts about the generation, evolution, and lifetime of QDA remain unknown (Knudsen et al., 2021, and references therein).

One interesting suggestion found in the literature is that "arc proper motion" (motion of an auroral arc relative to background plasma convection)  $\mathbf{u'} = \mathbf{u} - \mathbf{v}$ , where  $\mathbf{u}$  and  $\mathbf{v}$  are respectively arc velocity and plasma convection, may contain information about energy and momentum transfer between the magnetosphere and ionosphere (e.g., Kozlovsky et al., 2001; Haerendel et al., 1993). The meaning of arc proper motion is nonetheless debated (Borovsky et al., 2020, and references therein), at least partly due to a lack of estimates of  $\mathbf{u'}$  with sufficiently low uncertainty.

Here we envision using all-sky cameras to estimate  ${\bf u}$  and E3D to estimate  ${\bf v}$  at spatial and temporal resolutions exceeding those of previous studies (Haerendel et al., 1993; Frey et al., 1996; Williams et al., 1998; Kozlovsky et al., 2001). Typical in these studies were integration times of 10–60 s for one beam, and uncertainty in plasma drift velocity of 20–100 m/s depending on conditions, but mostly above 40 m/s, and displacement of the radar beam relative to the location of the arc by a few kilometers or tens of kilometers. Haerendel et al. (1993) also notes that the total displacement of the plasma during the observation window has an uncertainty of  $\pm 7$ –14 km. It could also be possible to obtain an additional, independent estimate of  ${\bf u}$  by for example tracking enhancements of E3D plasma parameter estimates.

In answer to the questions posed in the guidelines presented in Section 4.1 we arrive at the following.

- (i) QDA occurs over 60–80° MLat, with a peak occurrence rate at 70° MLat. It generally occurs between 14–08 MLT and is most frequently observed between 22 and 23 MLT (Syrjäsuo and Donovan, 2004), but can occur at virtually all MLTs.
   QDA arcs are generally aligned with geomagnetic east-west, with tilts of as much as ±8° locally (Gillies et al., 2014). Typical arc widths are 10–20 km (Aikio et al., 2002), with lifetimes of up to tens of minutes (Kozlovsky et al., 2001). QDA arcs have an emission height range of 80–400 km, with an emission peak at 110-km altitude (Davis, 1978).
- QDA is typically associated with Kp  $\leq$  4 (Karlsson et al., 2020). Given the range of MLats visible to E3D (Figure 1a) and the statistical MLat distribution of the auroral oval for Kp $\simeq$  2–4 (generally between 65–70° MLat; see Figures 2 and 4 in Carbary, 2005), QDA should be visible to E3D over  $\sim$ 22–05 MLT during both winter and summer. Note, however, that the probability of occurrence of QDA during summer relative to winter is decreased (e.g., Newell et al., 1996).
- (ii) Improving upon previous studies would ideally entail achieving a spatial resolution of a few hundred meters, or at most a few kilometers. The spatial extent of the E3D measurement volume normal to the arc would ideally span many arc widths (10–100 km) to accommodate variability in the location of the arc. We assume the longitudinal extent of the E3D measurement volume (i.e., spatial extent along the arc) is relatively less important for this experiment, since QDA arcs often extend longitudinally hundreds or even thousands of kilometers. A rough estimate of the measurement area at an altitude of 110 km is therefore a few to several hundred square kilometers.
- 320 (iii) Reported values of the magnitude of proper motion  $|\mathbf{u}'|$  lie within the range of approximately 20–150 m/s (Haerendel et al., 1993; Kozlovsky et al., 2001). The uncertainty of the plasma convection velocity in the direction normal to the arc would therefore ideally be roughly order of magnitude less than reported values of  $|\mathbf{u}'|$  say, 10 m/s or less. However, uncertainties of 25 m/s would likely be acceptable.
- (iv) Regarding possible beam patterns, we deem it desirable to be able to resolve gradients in convection velocity both along and across an auroral arc, and to minimize the uncertainty of convection velocity estimates given the geometry of the E3D sites. At the same time, quiet aurora are most frequently observed poleward of the three E3D sites. We therefore wish to select an observational area centered as close to the latitude at which QDA occurrence maximizes (i.e., 70° MLat) as possible without sacrificing measurement accuracy.
- Regarding bit length, to our knowledge it is not possible to analytically determine an optimal value. This choice was therefore made via trial and error, as we describe below.

Figure 3 illustrates an example "keogram" experiment designed to conform to the requirements given immediately above. Figure 3a shows that the experiment consists of 40 beams arranged in a rectangular  $(10\times4)$  grid. This beam pattern was selected via trial and error from several arbitrarily chosen beam patterns with varying numbers of beams. Our experimentation with different numbers of beams and beam patterns given a fixed observational area over E3D indicates that, for this particular experiment, the exact number of beams and the beam pattern itself is less important than selection of the overhead area and the total amount of time allotted for integrating over all beams, as the latter two are closely related to the overall level of uncertainty. This process also revealed that convection velocity uncertainty generally increases rapidly poleward of  $68^{\circ}$  MLat.

335

Figure 3. Keogram experiment design and resulting posterior convection uncertainties and resolution matrix for winter (second column) and summer (third column) conditions. **a:** Locations of observations made between 180- and 400-km altitude at intervals of 2.5 km (blue triangles), and their locations after mapping along geomagnetic field lines to 110-km altitude (black triangles). The extent of the observation region before and after mapping is indicated by the blue and gray boxes, respectively. The E3D sites are indicated by stars. **b:** The locations of the 1666 curl-free spherical elementary current system (SECS) poles used to derive the posterior convection velocity covariances. **c1–c2:** Square root of the diagonal of the posterior covariance matrix for east-west convection. **d1–d2:** Same as c1–c2, but for north-south convection. **e1–e2:** Resolution calculated following Madelaire et al. (2023).

For a wide variety of arbitrarily chosen beam patterns with total numbers of beams numbering between 20 and 120 we tested (not shown), we found that a total integration time of 300 s, or 7.5 s per beam, was necessary to achieve an acceptable level of plasma convection speed uncertainty ( $\sim$ 10–25 m/s) within the desired horizontal area of a few thousand square kilometers at 110-km altitude. (For reference, Stamm et al., 2021a, found that a per-beam integration time of 5 s was suitable for achieving uncertainties of 1–10 m/s, although it is critical to note that their model did not account for radar self-noise. When we exclude self-noise effects the uncertainties are reduced by factors of 4–6, not shown here.) A per-beam integration time of a few seconds is generally less than the integration times used in previous EISCAT systems, typically a minute or more. E3D allows for slightly shorter integration times as a result of the planned relatively higher transmission power of E3D (3.5 MW versus for example  $\sim$ 1 MW for the EISCAT Svalbard radar) since the ACF noise level  $\gamma$  is inversely proportional to the transmitter power  $P_T$  (cf. Equations 1–2, 4, 10).

We also found ideal range resolutions of 5 km and 2.5 km (corresponding to modulation bit lengths  $\Delta \tau = 16.7 \mu s$  and  $\Delta \tau = 33.4 \mu s$ ) for statistical ionospheric conditions, as given by IRI, during winter and summer respectively. Figure S1 in the Supplement shows a typical diagnostic figure used for making this assessment.

The plasma ACF is assumed sampled at regular intervals of 5 km (bit length  $\Delta \tau = 33.4 \mu s$ ) between 180- and 400-km altitude (blue triangles and box, Figure 3a), with ionospheric conditions specified using the International Reference Ionosphere-2016 (IRI2016) (Bilitza, 2018) and the Naval Research Laboratory-MSIS (Emmert et al., 2020) models for either "winter" (00:00 UT, December 23, 2020) or "summer" (00:00 UT, June 23, 2020) conditions. Results for winter and summer are respectively shown in the second and third columns of Figure 3. Our use of IRI2016 to specify the *F*-region ionospheric parameters that are the basis for the uncertainty calculations (including the uncertainty of ion drift velocity) is somewhat artificial, since IRI2016 does not account for the temperature and density enhancements that typically accompany auroral activity. We nonetheless deem the uncertainties we obtain from using IRI2016 to be a reasonable reference point.

At each measurement point, the resulting convection covariance matrix is then mapped down to 110-km altitude following the methodology of Reistad et al. (2024) (their equations 3 and 4). After mapping to a reference altitude of 110 km, the measurement points extend over  $\sim$ 66–68° MLat (black triangles and gray box in Figure 3a).

The convection covariance matrices are then used together with the grid of curl-free SECS (Vanhamäki and Juusola, 2020) shown in Figure 3b to obtain estimates of east-west and north-south convection velocity uncertainties (square root of the diagonal of the posterior convection covariance for each velocity component), as shown respectively in panels c1–c2 and d1–d2 of Figure 3. This is done by calculating the posterior model covariance matrix  $\Sigma_x$  via Equation 9, where the theory matrix  $\mathbf{A}$  now describes the relationship between the amplitudes of the SECS basis functions and the ionospheric convection pattern that they are used to reconstruct (see, e.g., Reistad et al., 2024; Laundal, 2022),  $\Sigma_m$  is the data covariance matrix (here the covariance of convection measurements mapped to 110-km altitude), and  $\Sigma_{x,pr}$  is the prior model covariance. (We use  $\Sigma_{x,pr}^{-1} = 0$ , i.e., no regularization.) The posterior convection covariance is then obtained as  $\Sigma_v = \mathbf{A}_v \Sigma_x \mathbf{A}_v^T$ , where  $\mathbf{A}_v$  is the theory matrix that maps SECS amplitudes to convection velocity components (equation 4 in Madelaire et al., 2023).

From these panels it is evident that the uncertainties are overall higher during winter than during summer within the experiment observation volume: typical uncertainties within the volume during winter conditions vary from 80 m/s to over 100 m/s but some places as low as 55 m/s, while during summer conditions the uncertainties are in some places lower than  $\sim$ 20 m/s.

Heuristically, for approximately equally spaced beams numbering between 20 and 120 within the area indicated by the blue box in Figure 3a, we find an order-of-magnitude average convection uncertainty  $\delta v_e \sim \delta v_n \sim \mathcal{O}\left(0.4B/\sqrt{T_{tot}}\right)$  m/s, where B is the area covered by the beam pattern at 110-km altitude in km<sup>2</sup> and  $T_{tot}$  is the total integration time for the entire beam pattern in seconds. For example, for  $B=2\times10^3$  km<sup>2</sup> and  $T_{tot}=300$  s, we have  $\delta v_e\sim\delta v_n\sim\mathcal{O}\left(50\right)$  m/s.

Results in Figures 3c1–d2 also indicate that the convection uncertainty is very high at latitudes poleward of  $\sim$ 67.3, regardless of season. This reflects the fact that the geometry of the three E3D sites is increasingly unfavorable for uniquely resolving all three convection velocity components as the radar beam is moved to higher latitudes.

Figures 3e1-e2 show the resolution as defined in section 2.2 of Madelaire et al. (2023) using point spread functions of the resolution matrix  $R = \Sigma_x A^T \Sigma_m^{-1} A$  (Madelaire et al., 2023, their equation 5). The resolution may be thought of as indicating the smallest resolvable scale size as a function of the locations and covariances of the convection measurements, and the locations of the SECS poles. It is clear that the smallest resolvable scale size within the gray box that defines the measurement volume is generally close to the spacing of the SECS poles (10 km and 5 km in the latitudinal and longitudinal directions, respectively). A spatial resolution of 5–10 km is acceptable for the plasma convection in producing estimates of arc proper motion  $\mathbf{u}'$ .

# 5.2 Example experiment design #2: Reconstruction of ionospheric convection







Rather than targeting a specific phenomenon or set of conditions, the goal of this experiment is to provide routine estimates of the high-latitude ionospheric convection pattern in the vicinity of E3D. As demonstrated by Reistad et al. (2024), measurements from such an experiment may enable a three-dimensional, volumetric reconstruction of the ionospheric current system, or estimation of the neutral wind over altitudes at which ion motion is dominated by collisions (typically below about 130-km altitude), or both. In answer to the questions posed in the guidelines presented in Section 4.1 we arrive at the following.

- (i) This experiment focuses on routine measurement of ionospheric convection and subsequent reconstruction of ionospheric electrodynamics rather than a particular ionospheric phenomenon. At the small-scale end, ionospheric convection can vary over distances of meters and times of seconds (Ivarsen et al., 2024; Huyghebaert et al., 2025, and references therein), while on the largest scales temporal and spatial variations occur over tens of minutes or hours and hundreds of kilometers, respectively (e.g., Jayachandran and MacDougall, 2007; Gillies et al., 2012). Here we choose to focus on what might be termed mesoscales, corresponding to temporal variations of order minutes and spatial variations of order a few kilometers. As Reistad et al. (2024) have shown via the E3D-based OSSE they present, it will likely be reasonable for an appropriately designed E3D experiment to resolve convection on these scales over an area of ~10<sup>5</sup> km<sup>2</sup>.
- (ii) The primary requirement for this experiment is that the spatial and temporal resolution be high enough to achieve the goal of the experiment, which is routine estimation of ionospheric convection and, insofar as possible, neutral winds

Figure 4. Reconstruction of ionospheric potential for three different beam patterns (from left to right): the Reistad et al. (2024) beam pattern, and 25- and 47-beam patterns covering respectively  $\sim 160 \text{ km}^2$  and  $\sim 180 \text{ km}^2$ . Details of each quantity are described in the main text. **a-c:** Original and reconstructed ionospheric potential patterns (thick gray lines and thin blue lines, respectively). The beam pattern at 120-km altitude is shown as orange dots in each panel. The E3D sites are indicated by yellow stars. **d-f:** Measurements of ionospheric convection along each beam with noise added (orange arrows) and the final reconstructed ionospheric convection pattern (black arrows). The green box indicates the region within which convection derived from the true and reconstructed potential pattern is sampled and shown in panels g-i. **g-i:** Reconstructed convection plotted against true convection (y and x axes, respectively). **j-i:** Square root of posterior prediction variance of eastward convection. **m-o:** Same as j-l, but for northward composite, **p-r:** Resolution defined according to Madelaire et al. (2023).

and volumetric ionospheric currents. As previously mentioned, this translates to scale sizes of a few kilometers and temporal scales of a few minutes. (For comparison, The typical spatial resolution of widely used Super Dual Auroral Radar Network, or SuperDARN, coherent radar estimates of ionospheric convection is of order 40–50 km; see Gjerloev et al., 2018, and references therein.)

(iii) Results from Reistad et al. (2024) indicate that resolving variations in ionospheric current densities with magnitudes of  $\sim$ 1–20  $\mu$ A/m<sup>2</sup> requires uncertainties of ionospheric convection estimates of no more than several tens of m/s. They report corresponding uncertainties in neutral wind components of 5–100 m/s, with the smallest uncertainties located near the center of the beam pattern.




(iv) The beam pattern presented by Reistad et al. (2024) (orange dots in Figure 4a) already meets the requirements for this experiment, and is therefore taken as a reference point. As with the previous example in Section 5.1, here we tested many arbitrarily selected beam patterns. For the vast majority of these, the resulting convection uncertainties were similar to those achieved by the Reistad et al. (2024) beam pattern. A total integration time of 600 s was used for all beam patterns. A range resolution  $\Delta r = 4$  km was selected via trial and error, although results for this experiment did not show any strong dependence on  $\Delta r$  between 2.5 km and 10 km. An alternative strategy for selecting range resolution, not employed here, is a range resolution diagnostic figure such as Figure S1 in the Supplement.

Following Reistad et al. (2024), we use output from the Geospace Environmental Model of Ion-Neutral Interactions (GEM-INI) as ground truth in the E3D example experiment presented in this section. We use the same modified E3D station locations that Reistad et al. (2024) used: each site is artificially moved 200 km south of its actual location "to probe a more relevant part of the GEMINI simulation output, covering the transition between the up and down FAC regions."

Figure 4 shows metrics for three different beam patterns, with each column corresponding to one beam pattern. The beam patterns at 120-km altitude are indicated with orange dots in panels a–c. The beam pattern shown in panel a is the pattern used for the reconstruction performed by Reistad et al. (2024) that is shown in their Figures 2 and 3. Each of the three reconstructions uses the same input potential pattern from GEMINI (blue contour lines in panels a–c) and system of spherical elementary currents. The beam pattern in the center column consists of 25 beams spaced by 40 km on a cubed sphere grid (area of  $\sim$ 160 km<sup>2</sup>). The beam pattern in the right column consists of 47 beams spaced by 30 km on a cubed sphere grid (area of  $\sim$ 180 km<sup>2</sup>).

Figures 4a-i show that differences in the three beam patterns play little role overall in reconstruction of the potential pattern (gray lines in panels a-c), and the plasma convection (black arrows in panels d-f and scatter points in panels g-i). (Two-dimensional histograms for each of the original and reconstructed convection components are also shown in Figure S2 of the Supplement.) Particularly in the vicinity of measurement-dense areas there appears to be little to no difference in the reconstruction. Correspondingly, Figures 4j-o show that the uncertainty (square root of diagonal elements of the covariance matrix) of the reconstructed eastward (j-l) and northward (m-o) components of convection tend to be lowest in the vicinity of measurements regardless of the overall beam pattern. That is, there is a clear connection between the density of measurements and

the overall reconstructed convection uncertainty. There is also an overall tendency for uncertainties of the eastward convection to be slightly lower than those of the northward convection at corresponding locations.

Figures 4p–r, which show the resolution as defined by Madelaire et al. (2023), indicate that the resolvable scale size at each location is closely tied to the density of measurements in the vicinity. It is clear that the third beam pattern (right column), in which the beam spacing is tightest, produces the most uniform resolution within the observational area. More variation in the resolution is seen in the first and second beam pattern; the second pattern in particular exhibits large gradients in resolution over much of the observational area, where resolutions in neighboring grid cells can in some locations differ by nearly 50 km.

We therefore conclude that for this particular experiment, which focuses on reconstruction of the ionospheric potential and the plasma convection over E3D, differences in beam patterns have somewhat surprisingly little influence on the overall reconstruction. Unless more specific information were given about the scientific requirements for this experiment, from the evidence in Figure 4 there is no clear reason to prefer one beam pattern over another. One exception is that it might in some cases be preferable to optimize the experiment for uniformity in the resolution and plasma convection uncertainties; in such cases the third, tightly spaced beam pattern with more uniform coverage may have some advantage.

## 450 6 Discussion and conclusions




In this study we have presented a framework for designing E3D experiments. This framework is derived from basic radar theory and incoherent scatter theory together with a number of simplifying assumptions.

We have produced an open-source implementation of this framework, *e3doubt* (Hatch and Virtanen, 2024), which is intended to greatly lower the barrier for experiment design. The goal is to enable researchers who are interested in using E3D or other ISR systems, but who are otherwise not experts, to get a feel for the possibilities and limitations of E3D, and achieve a basic level of independence and ownership for candidate E3D experiments. This framework enables relatively rapid testing of a number of experiment designs with minimal effort and computational overhead. We find in our tests that parameter uncertainty estimates for roughly 500 measurement points can be processed in a wall time of one minute on a modern laptop with 16 GB of ram and several CPUs.

As discussed in Section 2 our framework is based on a number of simplifying assumptions, including that the radar beam and pulse shapes are Gaussian, that individual pulse lengths can be ignored, and that the electron density profile in the ionosphere is Gaussian. (The latter assumption applies only to radar self-noise estimation.) We now expand briefly on each of these assumptions.

The assumption of a Gaussian beam shape and bit shape affects the shape of the scattering volume. This has little effect on the resulting uncertainty estimates, because we also assume that plasma parameters are constant within each scattering volume. As a practical example, Virtanen et al. (2014) made successful use of this assumption to calculate scattering volumes in analysis of real ISR data.

The assumption of a Gaussian pulse shape is made to enable fast computation of the beam intersection volumes. The effect of this assumption on the resulting uncertainty estimates is minimal, because the pulse shapes we use are always much longer than the modulation bit length.

The assumption that we may ignore the pulse length is equivalent to assuming that the experiment is well designed and we can sample the ACF to sufficiently long lags. This is justified in the F region, but may be sometimes optimistic in the E and D regions. This might partly explain why e3doubt uncertainty estimates in Figures 2e-h are lower than the sample standard deviations.

In the estimation of self-noise, the thickness of the (assumed Gaussian) electron density layer has a strong influence on the uncertainty estimates in high SNR conditions: Assuming a layer thickness that is in reality too narrow leads to underestimation of the plasma parameter uncertainties, while assuming a layer thickness that is too wide leads to overestimation of the uncertainties. This is yet another possible reason for the deviations between the GUISDAP error estimates and those modeled with e3doubt. On the other hand, we deem extremely accurate self-noise modeling to be of limited value in designing an experiment, since the values used in e3doubt will often be nothing more than (possibly educated) guesses about what the densities might be in reality.

An important indicator that the foregoing assumptions are reasonable are the validation results presented in Section 3 and Figure 2, which show that e3doubt uncertainty estimates are comparable to uncertainties obtained using the standard EISCAT analysis suite, GUISDAP, and to direct estimates of uncertainties via the standard deviation of measured plasma parameters.

As mentioned in the introduction, the effects of pulse lengths, self-noise, etc., can be modeled more accurately by means of creating synthetic voltage level radar signals, which are then decoded to ACFs and analysed just like real radar data. Much more technical open-source tools for conducting this type of simulation already exist for both E3D specifically (Virtanen et al., 2025) and more general radar systems (Swoboda et al., 2017). While clearly valuable for many types of analysis and experiment design, the added complexity is time consuming for testing experimental designs that have a large design parameter space.

485

490

We have presented two example experiments: one focused on reconstruction of ionospheric electrodynamics around an auroral arc, and another focused on understanding how beam selection can affect measurements from an E3D experiment. These examples are readily modifiable and expandable from the online code repository (Hatch and Virtanen, 2024). These examples take only a few minutes to run on a modern laptop.

Regarding expansion of e3doubt, a potential future improvement is to make an "expert version" that allows one to test different combinations of fitted parameters, prior models, and pulse lengths. This could include adding estimates of the plasma density from plasma lines via model priors.

Code and data availability. The code used to generate all of the figures shown in this study is freely available online (Hatch and Virtanen, 2024).

## **Appendix A: Incoherent scatter spectrum**

The ISgeometry package (Virtanen, 2023) is based on the incoherent scatter spectrum given as Equation 20 in Swartz and Farley (1979) for two ion species having masses of 30.5 amu (representing a mixture of  $O_2^+$  and  $NO^+$ ) and 16 amu ( $O^+$ ):

$$S(\mathbf{k}, \omega; N_e, T_e, T_i, \mathbf{V}_e, \mathbf{V}_i) = \frac{N_e r_e^2 \sin \delta}{\pi} \left( \left| \sum_j \mu_j y_j + i k^2 \lambda_D^2 \right|^2 \frac{\operatorname{Re}[y_e]}{\omega - \mathbf{k} \cdot \mathbf{V}_e} + |y_e|^2 \sum_j \frac{\eta_j \operatorname{Re}[y_j]}{\omega - \mathbf{k} \cdot \mathbf{V}_j} \right) \left( \left| y_e + \sum_j \mu_j y_j + i k^2 \lambda_D^2 \right|^2 \right)^{-1},$$
(A1)

where  $N_j, T_j$  and  $\mathbf{V}_j$  are the number density, temperature, and drift velocity of the  $j^{\text{th}}$  species,  $r_e$  is the classical electron radius,  $\delta$  is a polarization angle, and  $\lambda_D$  is the electron Debye length. The  $y_j$  functions are themselves functions of the Faddeeva function w(z) (which is related to the so-called plasma dispersion function  $Z(z) = i\sqrt{\pi}w(z)$ ), and  $\eta_j = N_j q_j^2/N_e e^2$ ,  $\mu_j = \eta_j T_e/T_j$ .

## Appendix B: Velocity covariance matrix estimation

We use the following procedure to obtain the velocity covariance matrix from estimates of the line-of-sight velocity from multiple transmitter-receiver pairs. It is taken directly from Section B.2.4 of Lehtinen et al. (2014).

Given the true velocity vector  $\mathbf{v} = (v_x, v_y, v_z)^T$  and the scattering wave unit vector  $\hat{\mathbf{k}}_s = \frac{1}{\|\mathbf{k}_s\|} (k_{sx}, k_{sy}, k_{sz})^T$  at site s, the line-of-sight velocity at site s is

$$v_s = \frac{1}{\|\mathbf{k}_s\|} \left( v_x k_{sx} + v_y k_{sy} + v_z k_{sz} \right). \tag{B1}$$

With line-of-sight velocity estimates from all  $N_s$  radar transmitter-receiver pairs, we have

$$(v_1, v_2, \dots, v_N)^T = \mathbf{A}_n \cdot \mathbf{v} + (\epsilon_1, \epsilon_2, \dots, \epsilon_N)^T,$$
(B2)

515 where

$$\mathbf{A}_v = \begin{pmatrix} \hat{\mathbf{k}}_1 & \hat{\mathbf{k}}_2 & \dots & \hat{\mathbf{k}}_{N_s} \end{pmatrix}^T. \tag{B3}$$

and  $\epsilon_s$  is the uncertainty of the line-of-sight velocity estimate at site s. The corresponding a posteriori covariance matrix of the full velocity vector is

$$\Sigma_{nv} = \left(\mathbf{A}_n^T \mathbf{\Sigma}_n^{-1} \mathbf{A}_v\right)^{-1}.$$
 (B4)

520 Author contributions. SMH was responsible for producing an initial draft of the manuscript, and conceiving of and writing the Python frontend for e3doubt. IV wrote ISGeometry (the R-language backend of e3doubt), conceived of and produced the experimental validation

shown in Figure 2, and outlined the discussion. KML contributed to defining the study, testing e3doubt, editing the manuscript, and creating and organizing International Space Science Institute team 506; the idea for this study is an outgrowth of discussions during team meetings. HWT performed extensive testing and validation of e3doubt, and gave feedback and suggestions on the manuscript. JV, AS, and DRH all contributed to discussions of the design and goals of e3doubt at various meetings, and provided expert guidance, suggestions and feedback regarding radar theory and the manuscript generally. JCH contributed expertise on and wrote an initial draft of example experiment 2 in Section 5.1, made an initial version of Figure 3, and performed tests of the e3doubt code.

Competing interests. The authors declare no competing interests.

525

Acknowledgements. SMH would like to thank Michael Madelaire for valuable insight and suggestions regarding model resolution and Example experiment design #2. IGRF model values are obtained via the ppigrf Python package (Laundal, 2022), which is a Python implementation (not frontend) of the IGRF. SMH and JCH were funded by the Research Council of Norway (RCN) under contract 344061. IV was funded by the Research Council of Finland project 347796. KML was funded by the European Union (ERC, DynaMIT, 101086985) and the Research Council of Norway through grant 300844/F50. HWT's work was supported by the Research Council of Finland project 354521 and the Kvantum Institute of the University of Oulu (POLAR project). AS acknowledges funding from the RCN grant 326039. DH was funded through a UiT The Arctic University of Norway contribution to the EISCAT\_3D project funded by Research Council of Norway through research infrastructure grant 245683.

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
