# Peer review of "Toolkit for incoherent scatter radar experiment design and applications to EISCAT\_3D"

_EGUsphere, 2025_

## Author Comment (AC1)

**Reviewer #1 Evaluation:**

We thank the reviewer for their thorough evaluation and constructive feedback on our manuscript. Based on this, we propose several changes below that we believe address the reviewers' concerns and improve the manuscript. These changes are summarized in this letter, along with specific responses to the reviewers' comments. Below the reviewer's comments our response is shown in **bold**. Proposed modifications and / or additions to the manuscript are shown in *italics*.

Prior to the start of EISCAT_3D radar observations, this paper shows how the accuracy of the ionospheric potential reconstruction changes by varying the beam pattern of the EISCAT_3D radar. It deserves publication with some minor modifications, as this is a very important toolkit needed when designing experiments according to the scientific objectives of each user.

**Thank you! We have strived to bring the manuscript and our responses here in line with recommendations from both reviewers.**

Minor revisions:

Figures 4g-i: The scatter plots are used to show that the accuracy of the estimation results is good, but information on where the residuals are small is lost if only the scatter plots are used. It is therefore recommended that the scatterplot is replaced or added to a two-dimensional heatmap displaying the residuals of the estimates for GEMINI.

**We agree that the scatter plots make it difficult to see where the residuals are small. On the other hand, after trying various strategies we also find it difficult to visually indicate where the residuals are small without significantly expanding Figure 4. We propose to instead present these results as heatmaps in Figure S2, shown below, which will be located in a new Supplement along with the following text:**

> *Figure S2 shows heatmaps of original versus reconstructed eastward (d–f), northward (g–i), and upward (j–l) velocity components for the three beam patterns shown in Figure 4. These heatmaps indicate that for all beam patterns the reconstructed convection velocities are generally within a few tens of m/s of the original convection velocities. Differences between the various beam patterns are mostly negligible but nevertheless visible.*

> *Results in this figure reinforce the conclusion in the main manuscript that the goal of the second example experiment in Section 5.2 (reconstruction of the ionospheric convection pattern with as little overall residual error as possible) is approximately equally well achieved by all three beam patterns.*

[Figure]

**Proposed caption for new Figure S2 in supplement:** Reconstruction of ionospheric potential for the three different beam patterns shown in Figure 4 (from left to right): the Reistad et al (2024) beam pattern, and 25- and 47-beam patterns covering respectively ~160 km² and ~180 km².

a–c: Original and reconstructed ionospheric potential patterns (thick gray lines and thin blue lines, respectively), identical to those shown in Figures 4a–c in the main article.

d–f: Reconstructed eastward convection plotted against true convection y and x axes, respectively).

g–i: Reconstructed northward convection plotted against true convection.

j–l: Reconstructed upward convection plotted against true convection.

Equation (6): Definition of "N" should be added.

**We propose to change the sentence just prior to Equation 6 so that it now reads** "*Thus the set of N measurements of the ACF at time lag τ with scattering vector **k** is represented by the measurement vector […]*".

Equation (10): It is suggested that a more detailed explanation of formula conversions be added, in addition to citing references, to make it easier for the reader to understand.

**We agree that more details are needed here, thank you for pointing this out. We propose to rewrite a large portion of the text following Equation 10 to describe in detail how we use Equation 10 in practice. We also propose to add several details about the uncertainty estimation process, which we hope the reviewer will agree makes the actual procedure easier for the reader to understand.**

**The following image contains our proposed rewrite of this section.**

**2.3 Uncertainty estimation procedure**

At the highest level, our uncertainty estimation procedure consists of two steps: estimation of the ACF measurement error covariance matrix $\Sigma_m$, and plasma parameter uncertainty estimation via Equation 9.

In the first step we estimate the ACF measurement covariance matrix $\Sigma_m$ as follows. We assume that measurements of the ACF at different lags are uncorrelated and have the same variance $\sigma^2$, so that $\Sigma_m$ is an identity matrix multiplied by $\sigma^2$. With these assumptions, the ACF noise level may be defined as (Vallinkoski, 1988)

$$\gamma^2 = \frac{\sigma^2}{f(\mathbf{k}, 0; \mathbf{x})^2 N_\tau} = \frac{N_a}{N_\tau} \frac{1}{\mathrm{SNR}^2}, \tag{10}$$

with $N_a$ the number of averaged ACF samples, $N_\tau \approx 2\pi/(\omega_0 \Delta\tau)$ the approximate number of lags to the first zero of the ACF assuming zero plasma bulk flow, $\omega_0 = k\sqrt{2k_B T_i/m_i}$ the "ion angular Doppler frequency" (Vallinkoski, 1988) for scattering wave number $k$, and $\Delta\tau = 2\Delta r/c$ the lag spacing (sampling interval) of ACF measurements. The lag spacing is matched with the user-specified range resolution $\Delta r$. The number $N_a$ accounts for the number of samples averaged in phase-code decoding, as well as post-integration in range and time.

With this definition, we calculate the SNR and the ACF noise level at each site as described in Section 2.1 using the radar equation assuming a particular set of radar system parameters, and assuming Gaussian beam shapes and pulse shapes. (See Sections B.3–B.6 in Lehtinen et al., 2014, as well as the routine "`multistaticNoiseLevels`" in Virtanen, 2023.) The outputs from this step are the separate ACF noise levels $\gamma_s$ for each transmitter-receiver pair $s$.

In the second step (plasma parameter uncertainty estimation) we use the fact that uncertainty of the line-of-sight velocity $v_i$ is practically independent of the errors of the other parameters, and assume that all receivers see the same $n_e$, $T_e$, and $T_i$. The latter assumption is equivalent to assuming that $T_e$ and $T_i$ are isotropic. (In reality $T_e$ and $T_i$ may be neither isotropic nor equal; see discussion in, e.g., Virtanen et al., 2014, and references therein.) The uncertainty estimation itself can therefore be divided into two parts: calculation of (i) the variance of estimates of scalar parameters, which do not have a direction, and (ii) the covariance of ion drift velocity vector estimates.

To calculate (i) we simply sum information from all receivers (since information or precision is the inverse of variance) to get a combined ACF noise level for the full multistatic system: $\gamma = \left(\sum_s \gamma_s^{-2}\right)^{-1/2}$. We then calculate the theoretical ACF $f(\mathbf{k}, \tau = 0; \mathbf{x})$, and solve for the ACF variance $\sigma^2$ in Equation 10 (see routines "`parameterFitErrors`" and "`ISspectrumSimple`" in Virtanen, 2023). We thus have an estimate of the ACF covariance matrix $\Sigma_m$. We then calculate the theory matrix $\mathbf{A}$, and finally calculate the posterior model covariance matrix in Equation 9, with the scalar plasma parameter uncertainties taken to be the square roots of its diagonal elements.

To calculate (ii) one must consider the system geometry, and in general the estimate of the line-of-sight ion drift for each transmitter-receiver pair has a different uncertainty. We combine the different uncertainties following the methodology outlined in Appendix B. For a monostatic system the ion drift velocity covariance matrix reduces to a scalar representing the variance of the line-of-sight ion velocity.

Line 173: It is helpful for readers to add more detailed explanation about B.2.4 of Lehtinen et al. (2014).

**We propose to include in the revised manuscript a new Appendix B that gives a full description of the procedure we use. The text of the proposed Appendix B is contained in the following image:**

**Appendix B: Velocity covariance matrix estimation**

485    We use the following procedure to obtain the velocity covariance matrix from estimates of the line-of-sight velocity from multiple transmitter-receiver pairs. It is taken directly from Section B.2.4 of Lehtinen et al. (2014).

Given the true velocity vector $\mathbf{v} = (v_x, v_y, v_z)^T$ and the scattering wave unit vector $\hat{\mathbf{k}}_s = \frac{1}{\|\mathbf{k}_s\|}(k_{sx}, k_{sy}, k_{sz})^T$ at site $s$, the line-of-sight velocity at site $s$ is

$$v_s = \frac{1}{\|\mathbf{k}_s\|}(v_x k_{sx} + v_y k_{sy} + v_z k_{sz}). \tag{B1}$$

490    With line-of-sight velocity estimates from all $N_s$ radar transmitter-receiver pairs, we have

$$(v_1, v_2, \ldots, v_{N_s})^T = \mathbf{A}_v \cdot \mathbf{v} + (\epsilon_1, \epsilon_2, \ldots, \epsilon_{N_s})^T, \tag{B2}$$

where

$$\mathbf{A}_v = \begin{pmatrix} \hat{\mathbf{k}}_1 & \hat{\mathbf{k}}_2 & \ldots & \hat{\mathbf{k}}_{N_s} \end{pmatrix}^T. \tag{B3}$$

20

and $\epsilon_s$ is the uncertainty of the line-of-sight velocity estimate at site $s$. The corresponding a posteriori covariance matrix of the
495    full velocity vector is

$$\Sigma_{pv} = \left(\mathbf{A}_v^T \Sigma_v^{-1} \mathbf{A}_v\right)^{-1}. \tag{B4}$$

**Line 201–202: Why were uncertainties estimated by this study underestimated relative to those of GUISDAP above ~300-km altitudes? Are assumptions used in GUISDAP desirable compared to e3doubt?**

This is a reasonable question that is unfortunately not easy to answer. Our combined experience with GUISDAP is that the code is opaque, and documentation of the exact assumptions that GUISDAP makes is, to our knowledge, nowhere publicly available.

Two points that may be worth raising are the following.

1. Both GUISDAP and our uncertainty estimation procedure (e3doubt) allow for variable range resolution as a function of altitude. For this particular experiment the range resolution was generally less than 5 km below 120-km altitude, less than 10 km below 160-km altitude, and between 10 and 30 km above 160-km altitude. Uncertainty decreases with decreasing spatial resolution (i.e., larger range resolution).

2. GUISDAP has an option for modeling error correlations that increases the computational cost of the calculations by a factor of approximately $10^4$, making use of this option impractical for the vast majority of analysis. However, one of us (IV) has investigated and found that inclusion of error correlation effects quite precisely creates the approximate factor-of-two difference between GUISDAP uncertainty estimates and the sample standard deviations shown in Figure 2.

In the revised manuscript we can make the point about variable range resolution and slightly expand the discussion of error correlation if the reviewer agrees that these points are informative.

Line 314: The abbreviation "SECS" should be added after "spherical elementary current system".
Line 331: "Madelaire et al. (2023)" should be "(Madelaire et al., 2023)".
Line 345: "(Reistad et al., 2024)" should be "Reistad et al. (2024)".

**We will correct all of these in the revised manuscript. Thank you for catching these mistakes.**

---

## Author Comment (AC2)

This paper presents a software toolkit ("e3doubt") that allows to estimate the uncertainty of plasma parameter measurements with the upcoming EISCAT_3D incoherent scatter radar system. Since phased-array incoherent scatter radars like EISCAT_3D can be run in a large variety of measurement settings to accommodate different spatial and temporal resolutions, such a toolkit will allow conducting individual observing system simulation experiments (OSSEs) for specific processes. The low computational requirements and its applicability for ionosphere scientists who are not ISR experts make e3doubt a clear improvement over existing software.

The underlying equations (e.g., radar equation, incoherent scatter spectrum, noise levels) are well described, and the applied assumptions and simplifications are clearly stated. Two experiment examples are presented to demonstrate the developed toolkit. The uncertainties estimated with e3doubt for a 1-hour measurement window with the existing EISCAT UHF are compared to uncertainty estimates given by the ISR analysis software GUISDAP. It is shown that the e3doubt uncertainty estimate is very close to the GUISDAP estimate, though both are considerably lower than the actual variability of the plasma parameters.

There are some minor concerns regarding the provided guidelines in Section 4. If this section is to be seen as a general manual for e3doubt, a more distinguished identification of the single steps should be provided during the examples in Section 5. Additionally, I think it would be beneficial if the examples in Section 5 were more focused on how e3doubt can help with the decision-making process when designing E3D experiments. Other open questions remain about the underestimation of parameter uncertainties in the topside ionosphere (Figure 2) and a more detailed discussion of the simplifying assumptions that are employed to allow for the low computational overhead in comparison with Swoboda et al., 2017.

Overall, the paper is well-written and addresses an important issue. The comments below mostly address the discussion and presentation of the provided examples. I therefore see the paper to be suitable for publication after minor revisions.

We thank the reviewer for their detailed review of the manuscript, and for their encouraging feedback. The proposed revisions below represent our best attempt to bring the presentation in line with suggestions from both reviewers. Below each comment from the reviewer our response is shown in **bold**. Proposed modifications and / or additions to the manuscript are shown in *italics*.

Minor comments:

**Guidelines in Section 5 examples**

Section 4 introduces a set of guidelines for designing E3D experiments to study specific processes with the help of e3doubt. I think this is an excellent approach, but the demonstration in Section 5 is somewhat incomprehensible. In Section 5.1, steps 1-3 of the proposed guidelines are described in detail, but step 4 is neglected, though it is the crucial part of the whole process. The selection of the beam pattern shown in Figure 3 appears to be somewhat arbitrary. Why was this beam pattern selected? How does the sampling in 5 km intervals translate to bit length? Why was an integration time of 7.5s chosen (see also in

 It would be helpful if step 4 of the guidelines were discussed similarly as the first three steps. Maybe a comparison of two (reasonable) experiment setups and their impact on the resulting errors would be helpful. In Section 5.2, a similar step-by-step explanation should be added.

**The reviewer here highlights several points that deserve addressing. We have chosen to break them down into separate bullet points in order to give a focused reply.**

- The selection of the beam pattern shown in Figure 3 appears to be somewhat arbitrary. Why was this beam pattern selected?

**The beam pattern was indeed arbitrarily chosen. Given that the focus of this hypothetical experiment is arc proper motion, our motivation is to cover as large a line in the "cross-arc" (typically north-south) direction as possible while still being able to resolve some minimal amount of variation along the arc. We propose adding the following text to the revised manuscript:**

*Regarding possible beam patterns, we deem it desirable to be able to resolve gradients in convection velocity both along and across an auroral arc, and to minimize the uncertainty of convection velocity estimates given the geometry of the E3D sites. At the same time, quiet aurora are most frequently observed poleward of the three E3D sites. We therefore wish to select an observational area centered as close to the latitude at which QDA occurrence maximizes (i.e., 70° MLat) as possible without sacrificing measurement accuracy.*

*[…]*

*Figure 3a shows that the experiment consists of 40 beams arranged in a rectangular (10×4) grid. This beam pattern was selected via trial and error from several arbitrarily chosen beam patterns with varying numbers of beams. Our experimentation with different numbers of beams and beam patterns given a fixed observational area over E3D indicates that, for this particular experiment, the exact number of beams and the beam pattern itself is less important than selection of the overhead area and the total amount of time allotted for integrating over all beams, as the latter two are closely related to the overall level of uncertainty. This process also revealed that convection velocity uncertainty generally increases rapidly poleward of 68° MLat.*

*For a wide variety of arbitrarily chosen beam patterns with total numbers of beams numbering between 20 and 120 we tested (not shown), we found that a total integration time of 300 s, or 7.5 s per beam, was necessary to achieve an acceptable level of plasma convection speed uncertainty within the desired horizontal area of a few thousand square kilometers at 110-km altitude.*

- How does the sampling in 5 km intervals translate to bit length?

**This translates to a bit length of 33.4 μs. We will include this information in the revised manuscript.**

- Why was an integration time of 7.5s chosen (see also in the next comment)?

**This is approximately the smallest per-beam integration time that we found we could use to achieve the desired overall level of uncertainty within the beam pattern, as**

**described in the proposed text revision above. In addition, we propose to add the following point of clarification on the question of per-beam integration time:**

*For reference, Stamm et al., 2021a, found that a per-beam integration time of 5 s was suitable for achieving uncertainties of 1–10 m/s, although it is critical to note that their model did not account for radar self-noise. When we exclude self-noise effects (not shown) the uncertainties are reduced by factors of 4–6. A per-beam integration time of a few seconds is generally less than the integration times used in previous EISCAT systems, typically a minute or more. E3D allows for slightly shorter integration times as a result of the planned relatively higher transmission power of E3D (3.5 MW versus for example ~1 MW for the EISCAT Svalbard radar) since the ACF noise level γ is inversely proportional to the transmitter power $P_T$ (cf. Equations 1–2, 4, 10).*

- It would be helpful if step 4 of the guidelines were discussed similarly as the first three steps.

**This is an excellent point, we will include a discussion of all four steps of the guidelines in both example experiment subsections.**

**For the first example experiment we propose the following four points. (For the second example experiment, the four points we propose are given later in this response.)**

1. *QDA occurs over 60–80° MLat, with a peak occurrence rate at 70° MLat. It generally occurs between 14–08 MLT and is most frequently observed between 22 and 23 MLT (Syrjäsuo and Donovan, 2004), but can occur at virtually all MLTs. QDA arcs are generally aligned with geomagnetic east-west, with tilts of as much as ±8° locally (Gillies et al., 2014). Typical arc widths are 10–20 km (Aikio et al., 2002), with lifetimes of up to tens of minutes (Kozlovsky et al., 2001). QDA arcs have an emission height range of 80–400 km, with an emission peak at 110-km altitude (Davis, 1978).*

   *QDA is typically associated with Kp ≤ 4 (Karlsson et al., 2020). Given the range of MLats visible to E3D (Figure 1a) and the statistical MLat distribution of the auroral oval for Kp≈ 2–4 (generally between 65–70° MLat; see Figures 2 and 4 in Carbary, 2005), QDA should be visible to E3D over ~22–05 MLT during both winter and summer. Note, however, that the probability of occurrence of QDA during summer relative to winter is decreased (e.g., Newell et al., 1996)*

2. *Improving upon previous studies would ideally entail achieving a spatial resolution of a few hundred meters, or at most a few kilometers. The spatial extent of the E3D measurement volume normal to the arc would ideally span many arc widths (10–100 km) to accommodate variability in the location of the arc. We assume the longitudinal extent of the E3D measurement volume (i.e., spatial extent along the arc) is relatively less important for this experiment, since QDA arcs often extend longitudinally hundreds or even thousands of kilometers. A rough estimate of the measurement area at an altitude of 110 km is therefore a few to several hundred square kilometers.*

3. *Reported values of the magnitude of proper motion |u'| lie within the range of approximately 20–150 m/s (Haerendel et al., 1993; Kozlovsky et al., 2001). The uncertainty of the plasma convection velocity in the direction normal to the arc would therefore ideally be roughly order of magnitude less than reported values of |**u'**| — say, 10 m/s or less. However, uncertainties of 25 m/s would likely be acceptable.*

4. *Regarding possible beam patterns, we deem it desirable to be able to resolve gradients in convection velocity both along and across an auroral arc, and to minimize the uncertainty of convection velocity estimates given the geometry of the E3D sites. At the same time, quiet auroras are most frequently observed poleward of the three E3D sites. We therefore wish to select an observational area centered as close to the latitude at which QDA occurrence maximizes (i.e., 70° MLat) as possible without sacrificing measurement accuracy.*

   *Regarding bit length, to our knowledge it is not possible to analytically determine an optimal value. This choice was therefore made via trial and error, as we describe below.*

- Maybe a comparison of two (reasonable) experiment setups and their impact on the resulting errors would be helpful.

**Thank you for pointing out, here and elsewhere, that there was insufficient discussion of the choice of beam pattern in the original manuscript.**

**To arrive at the beam pattern we show in Figure 3, we in fact tested many (several tens of) arbitrarily selected beam patterns. Instead of showing two arbitrarily selected beam patterns, we propose to greatly expand the discussion of the choice of beam pattern in Section 5.1. We propose that this expanded discussion include the following paragraph:**

*Our experimentation with different numbers of beams and beam patterns given a fixed observational area over E3D also indicates that, for this particular experiment, the exact number of beams and beam pattern is less important than selection of the overhead area and the total amount of time allotted for integrating over all beams, as the latter two are closely related to the overall level of uncertainty. In particular, for a wide variety of arbitrarily chosen beam patterns with total numbers of beams numbering between 20 and 120 we tested (not shown), we find that total integration times of at least hundreds of seconds are necessary to achieve the desired level of plasma convection speed uncertainty (~10–25 m/s) within the desired horizontal area (a few thousand square kilometers at 110-km altitude).*

- In Section 5.2, a similar step-by-step explanation should be added.

**We propose to add the following step-by-step explanation to Section 5.2.**

*Rather than targeting a specific phenomenon or set of conditions, the goal of this experiment is to provide routine estimates of the high-latitude ionospheric convection pattern in the*

*vicinity of E3D. As demonstrated by Reistad et al. (2024), measurements from such an experiment may enable a three-dimensional, volumetric reconstruction of the ionospheric current system, or estimation of the neutral wind over altitudes at which ion motion is dominated by collisions (typically below about 130-km altitude), or both. In answer to the questions posed in the guidelines presented in Section 4.1 we arrive at the following.*

1. *This experiment focuses on routine measurement of ionospheric convection and subsequent reconstruction of ionospheric electrodynamics rather than a particular ionospheric phenomenon. At the small-scale end, ionospheric convection can vary over distances of meters and times of seconds (Ivarsen et al., 2024; Huyghebaert et al., 2025, and references therein), while on the largest scales temporal and spatial variations occur over tens of minutes or hours and hundreds of kilometers, respectively (e.g., Jayachandran and MacDougall, 2007; Gillies et al., 2012). Here we choose to focus on what might be termed mesoscales, corresponding to temporal variations of order minutes and spatial variations of order a few kilometers. As Reistad et al. (2024) have shown via the E3D-based OSSE they present, it will likely be reasonable for an appropriately designed E3D experiment to resolve convection on these scales over an area of $\sim 10^5\, km^2$.*

2. *The primary requirement for this experiment is that the spatial and temporal resolution be high enough to achieve the goal of the experiment, which is routine estimation of ionospheric convection and, insofar as possible, neutral winds and volumetric ionospheric currents. As previously mentioned, this translates to scale sizes of a few kilometers and temporal scales of a few minutes. (For comparison, The typical spatial resolution of widely used Super Dual Auroral Radar Network, or SuperDARN, coherent radar estimates of ionospheric convection is of order 40–50 km; see Gjerloev et al., 2018, and references therein.)*

3. *Results from Reistad et al. (2024) indicate that resolving variations in ionospheric current densities with magnitudes of $\sim 1$–$20\ \mu A/m^2$ requires uncertainties of ionospheric convection estimates of no more than several tens of m/s. They report corresponding uncertainties in neutral wind components of 5–100 m/s, with the smallest uncertainties located near the center of the beam pattern.*

4. *The beam pattern presented by Reistad et al. (2024) (orange dots in Figure 4a) already meets the requirements for this experiment, and is therefore taken as a reference point. As with the previous example in Section 5.1, here we tested many arbitrarily selected beam patterns. For the vast majority of these, the resulting convection uncertainties were similar to those achieved by the Reistad et al. (2024) beam pattern. A total integration time of 600 s was used for all beam patterns. A range resolution $\Delta r = 4\ km$ was selected via trial and error, although results for this experiment did not show any strong dependence on $\Delta r$ between 2.5 km and 10 km. An alternative strategy for selecting range resolution, not employed here, is a range resolution diagnostic figure such as Figure S1 in the Supplement.*

**The new diagnostic figure, Figure S1, that is referred to in the fourth point is shown below. The accompanying text that we propose is as follows.**

*Figure S1 shows a typical diagnostic figure used for selecting an ideal range resolution for the first example experiment presented in Section 5.1 of the main article. Panel a is identical to Figure 3a in the main text, except that the first beam is highlighted in orange. As in Figure 3 of the main text, the center column shows results for winter conditions and the right column results for summer conditions.*

*For winter conditions, Figure S1b1 shows the uncertainty of the magnetic eastward convection component as a function of altitude on the y axis as well as modulation bit length $\Delta r_{mod}$ (lower x axis) and lag spacing $\Delta\tau$ (upper x axis) for a constant range gate resolution of 15 km over 180-km to 400-km altitude. The uncertainty averaged over all altitudes is shown in Figure S1c1. The lowest average uncertainty as a function of altitude occurs for 5 km $\leqslant \Delta r_{mod} \leqslant$ 10 km.*

*Corresponding quantities for summer conditions are shown in Figures S1b2 and S1c2. From the latter figure it is clear that the average uncertainty minimizes near $\Delta r_{mod}$ = 2.5 km.*

*These results motivate our selection of $\Delta r_{mod}$ = 5 km for winter conditions and $\Delta r_{mod}$ = 2.5 km for summer conditions in Section 5.1 of the main text.*

[Figure]

**Caption for Figure S1 in proposed Supplement.** Example diagnostic figure used for selecting range resolution values in the first example experiment, with results for winter and summer conditions shown in the center and right columns, respectively. **a:** Beam pattern layout in the same format as Figure 3a of the main article, with locations of observations made between 180- and 400-km altitude at intervals of 15 km shown as blue triangles, and their locations after mapping along geomagnetic field lines to 110-km altitude as black triangles. The extent of the observation region before and after mapping is indicated by the blue and gray boxes, respectively. **b1–b2:** Uncertainty of magnetic eastward convection as a function of altitude on the y axis as well as modulation bit length $\Delta r_{mod}$ (lower x axis) and lag spacing $\Delta\tau$ (upper x axis). **c1--c2:** Uncertainty of magnetic eastward convection averaged over all altitudes.

**Integration time**

In line 302, you state that the integration time per beam for the chosen experiment mode is 7.5 s. This is significantly shorter than the common integration times for the classical EISCAT systems (~1 min, sometimes 30s). Does EISCAT_3D generally allow for lower integration times (e.g., due to higher antenna gain, transmission power)? Please add a short clarification for readers who are familiar with the classical EISCAT systems but not the upcoming E3D.

**We address the choice of per-beam integration time and how E3D compares to previous EISCAT radar systems in our foregoing replies to reviewer comments and proposed revisions to the text. We hope the reviewer will agree that the revisions we propose above, which highlight that the E3D transmitter power is somewhat greater than the previous generation of EISCAT radars, and point out that a per-beam integration time of a few seconds was also used in the E3D-based study of Stamm et al (2021), will sufficiently address this point.**

**Pre-defined set-ups (Common Programmes)**

As mentioned already above, the selection of the beam patterns in Section 5 (except for the one taken from Reistad et al.) appears to be somewhat arbitrary. Does e3doubt contain a pre-defined set of beam patterns, and if yes, how (by what criteria) are they selected? Making the experiment design process more accessible is, from my point of view, the key point of the paper, and hence, should be explained rather than just stated. Once E3D Common Programmes have been designed and selected, will they be made available as predefined setups in e3doubt?

**The reviewer is exactly right, the beam patterns are arbitrary. We will attempt to make this point much clearer throughout the revised manuscript, as we have described in the proposed text revisions above. These revisions all aim to clarify that the beam patterns we present are arrived at via trial and error instead of a strict methodology or procedure.**

**Regarding possible future default programs / beam patterns, these could of course easily be included as defaults for e3doubt. Our hope is that in providing all of the scripts that generate the various figures (including the beam patterns they display) in the e3doubt repository, we have made it remarkably straightforward for an interested user to implement their own beam patterns since this requires nothing more than a list of beam elevations and azimuths. To clarify this point, we propose to modify the opening paragraph of Section 3 ("Implementation and validation") as follows:**

*We have produced an open-source toolkit consisting of a core implementation of the uncertainty estimation procedure described in Section 2 called ISgeometry (Virtanen, 2023), which is written in the R programming language, and a front-end called e3doubt (Hatch and Virtanen, 2024) that is written in Python. The latter includes scripts that reproduce the analysis described in this section and Section 5, as well as several additional tools. Exploration of different candidate beam patterns, for example, is straightforward and requires nothing more than a text-based list of beam pointing directions (elevations and azimuths).*

**Differences between e3doubt and GUISDAP uncertainties**

In Figure 2 e-h, it can be seen that in the topside ionosphere, the e3doubt uncertainty estimates are lower than the GUISDAP estimates for all plasma parameters. Can this trend be pinned to a specific assumption made in Section 2? Also, there is an interesting spike in the e3doubt uncertainty estimate for electron density around the F2 peak. I am aware that such features cannot always be explained in detail, but I would like to see the discussion of Figure 2 e-h in lines 191-198 extended.

**Regarding the first question about whether e3doubt uncertainty estimates being lower than those of GUISDAP can be attributed to a specific assumption made in Section 2, one possibility is self-noise modeling in e3doubt. Please see our reply below to the reviewer's comment on the assumptions we make and low computational demand.**

**We thank the reviewer for pointing out the spike in the e3doubt-based estimate of $n_e$ uncertainty. We will extend the discussion around the lines the reviewer has indicated by incorporating both this point and a few additional points regarding differences and similarities between e3doubt- and GUISDAP-based uncertainty estimates and the experimental sample standard deviations. We propose the following text.**

*The overall similarity in Figures 2e–h between the uncertainty estimates given by GUISDAP and our uncertainty estimation procedure (e3doubt) indicates that both produce reasonable, if optimistic, estimates of the uncertainties of plasma parameters derived from ISR measurements. Beyond this main observation, some additional observations are in order. First, comparison of GUISDAP and e3doubt uncertainties points to an apparent tendency for e3doubt to underestimate uncertainties relative to those of GUISDAP above ~300-km altitude. Whether this difference is related to one or more of the assumptions we make in Section 2 is unfortunately unclear. On the other hand, the sample standard deviations indicated by orange lines in Figures 2e–h have slight local maxima at altitudes between ~225–275 km (e.g., local peaks in std($n_e$) and std($T_e$) near 250-km altitude).*

**Discussion of assumptions for low computational demand**

One of the key improvements of the presented toolkit is its low computational demands and accessibility to non-radar experts. Hence, the discussion in lines 383-386 is rather short and could be extended. Why is it ok to make these assumptions, and what are the possible consequences on the uncertainty estimate? Especially, the assumptions of a Gaussian beam pattern and pulse shape (l. 106) and a Gaussian electron density profile (l. 114f) for the self-noise calculation are not straightforward to me and should be discussed in more detail.

Help! Here is the text from lines 380–386 for context

The simplifying assumptions that we employ facilitate relatively rapid testing of a number of experiment designs with minimal effort and computational overhead. We find in our tests that parameter uncertainty estimates for roughly 500 measurement points can be processed in a wall time of one minute on a modern laptop with 16 GB of ram and several CPUs. This simplified approach comes at the cost of neglecting (for example) pulse lengths and error covariances. The self-noise model is also very simplistic, and the antenna arrays are assumed to be horizontal. The effects of pulse lengths, self-noise, etc., can be modeled

accurately by means of creating synthetic voltage level radar signals, which are then decoded to ACFs and analysed just like real radar data. As mentioned in the introduction, much more technical open-source tools for conducting this type of simulation already exist (Swoboda et al., 2017). While clearly valuable for some types of analysis, the added complexity is time consuming for testing experimental designs that have a large design parameter space.

**Here the reviewer starts by asking why it is OK to make the assumptions that we do, and what are the consequences for the uncertainty estimate? The simplest but least satisfying answer to the question of why it is OK is that results in Figure 2 indicate that the assumptions we make lead to reasonable estimates of plasma parameter uncertainties.**

**To discuss the question of what consequences these assumptions have, we reframe the rest of the reviewer's comment into three separate questions which we then answer below: (i) What are the effects of assuming a Gaussian beam shape? (ii) Assuming a Gaussian pulse shape? (iii) Ignoring the pulse length? (iv) Assuming a Gaussian electron density profile in self-noise estimation?**

**(i) Gaussian beam shapes and bit shapes affect shapes of the scattering volumes. However, this does not have much effect on the final results, because we assume that plasma parameters are constant within each scattering volume. As a practical example, Virtanen et al. (2014) made successful use of the same approximation to calculate scattering volumes in analysis of real ISR data.**

**(ii) The Gaussian pulse shapes are used just to enable fast computation of the beam intersection volumes. Their effect on the results should be minimal, because the pulse shapes we use are always much longer than the modulation bit length.**

**(iii) Ignoring the pulse length is equivalent to assuming that the experiment is well designed and we can sample the ACF to sufficiently long lags. This is justified in the _F_ region, but may be a bit optimistic in D and E regions. This might partly explain why e3doubt uncertainty estimates in Figures 2e–h are lower than the sample standard deviations.**

**(iv) Our method for self-noise estimation is rather simplistic, and changing thickness of the Gaussian $n_e$ layer has a strong impact on the error estimates in high SNR conditions. Assuming a layer thickness that is in reality too narrow leads to underestimation of the plasma parameter uncertainties, while assuming a layer thickness that is too wide leads to overestimation of the uncertainties. This is one possible reason for the deviations between the GUISDAP error estimates and those modelled with e3doubt. On the other hand, we do not see the point in extremely accurate self-noise modeling, because plasma density $n_e$ has a huge influence on the uncertainties, and the values we use in e3doubt are just our (educated) guesses about what the densities might be in reality.**

**If the reviewer agrees that our response here satisfactorily answers the questions they have posed, we propose to extend the discussion around the lines indicated by the reviewer based on our response.**

Technical comments:

Line 1 (and later): "phased-array" and "phased array" should be used consistently (I think no hyphen is more common, but both are acceptable)

**We agree, this will be corrected in the revision. In our usage in this manuscript we use "phased-array" with a hyphen to indicate that it is acting as an adjective.**

Line 5: "the the"

**Thank you, we will correct this in the revision.**

Line 11: Technically, the mean molecular mass also affects the spectrum. As stated later in the paper, 30.5u is usually assumed, which is a fair estimate. I leave it to the authors if they want to add the mean molecular mass to this list

**This is a good idea, we will add mean molecular mass to the list in the revision.**

Line 16: The duty cycle is not discussed later in the paper (e.g., in Section 4 together with bit length and range resolution).

**Thank you for catching this. We propose to make explicit in the revised manuscript that we assume the reader has basic familiarity with concepts such as duty cycle, bit length, and beam width, and that only the latter two are defined and discussed in Section 4.1.**

Figure 1c: The field strength values are cut off at the edge of the figure

**Thank you for pointing this out, this will be corrected in the revised manuscript.**

Line 197f: the ACF covariance matrix $\Sigma l$ is not labeled as such in Section 2. Is it equivalent to $\Sigma m$?

**Thank you for catching this misprint. This indeed should have been $\Sigma_m$ and will be corrected in the revision.**

Line 402 and Equation A1: "the kth species". From Equation A1, I would think the labelling of ion species is j?

**This will be corrected, thank you for catching this.**